# "I just felt that everything came tumbling down around me"—Barriers in cancer care for patients with severe mental illness: A qualitative study

**Astrid Næraa Høeg Vendelsøe**[1]*, **Mette Stie**[1,2], **Peter Hjorth**[2,3], **Jens Søndergaard**[4], **Dorte Gilså Hansen**[4], **Lars Henrik Jensen**[1,2]

1 Department of Oncology, Vejle Hospital, University Hospital of Southern Denmark, Odense, Denmark, 2 Department of Regional Health Research, Faculty of Health Sciences, University of Southern Denmark, Odense, Denmark, 3 Psychiatric Department, Vejle Mental Health Services, Region of Southern Denmark, University Hospital of Southern Denmark, Odense, Denmark, 4 The Research Unit for General Practice, Department of Public Health, University of Southern Denmark, Odense, Denmark

* a.n@rsyd.dk

**Data Availability Statement:** This manuscript and its supplementary materials provide sufficient

## Abstract

### Background

Patients with severe mental illness experience serious inequity when facing cancer treatment. They are less likely to be referred for cancer treatment following recommended guidelines and have poorer cancer survival than patients without mental illness. Relevant specialties such as psychiatry and general practice are rarely involved, and the patient perspective is rarely represented in research in the field. The present study investigated how patients with severe mental illness experience barriers to and facilitators of patient-centred cancer treatment and care.

### Methods

In this qualitative case study, field observations, semi-structured interviews, and patient file analysis were performed with five patients with cancer in an adult psychiatric setting, included through purposeful sampling.

### Results

Our analysis showed one major theme, "Complexity on many levels", and four subthemes: "How the mental illness is affected by the cancer trajectory", "The complexity of patient vulnerability", "Fragmented healthcare system and lack of structure", and "The role of the relationship between patient and health professional." Barriers included the cancer trajectory leading to severe worsening of the mental illness, as well as fragmentation of the healthcare system and a lack of a systematic approach to the patient group. Facilitators included the health professionals acknowledging the patient's own resources and approaching the patient as a person rather than a disease.

details to reproduce the study, including individual, de-identified examples from patient interviews and observations. However, the full interview transcripts, conducted in Danish, contain identifiable and sensitive information and cannot be shared in full, as participants did not consent to the publication of their entire transcripts. This decision aligns with the data ethical approval obtained from the Region of Southern Denmark. Access to the data may be granted to qualified researchers who meet the criteria for access to confidential information, upon request to the corresponding author (contact via lars.henrik.jensen@rsyd.dk), and in compliance with legal and ethical standards governing data privacy.

**Funding:** The study was funded by the Novo Nordisk Foundation, https://novonordiskfonden.dk/en/. Originally the grant (grant number NNF21OC0069961) was awarded to Dorte Gilså Hansen (DGH), who then transferred the main responsibility of the project - and therefore the grant - to LHJ. The funders had no role in study design, data collection and analysis, decision to publish, or preparation of the manuscript.

## Conclusion

This study highlights critical focal points to improve care for patients with cancer who also struggle with severe mental illness. By addressing these target areas, healthcare providers can better tailor their approach to meet the unique needs of this population.

## Introduction

Patients with and without mental illness have the same cancer incidence, but cancer survival is significantly lower in patients with simultaneous mental illness [1–4]. In the case of severe mental illness such as schizophrenia and bipolar disorder, patients face two to threefold higher mortality rates corresponding to a shorter life expectancy of 10–20 years [3, 5, 6]. This disparity has often been attributed to somatic multi-morbidity, adverse effects of antipsychotic medication, poor lifestyle choices and late stage cancer at the time of diagnosis [5, 7, 8]. However, recent studies have shown increased mortality even when cancer stage is adjusted for [9–12]. Part of the disparity might then be explained by the fact that the patients are less likely to receive treatment following national guidelines [4, 11, 12]. The literature has uncovered other barriers pointing to a serious inequity, defined by the World Health Organization as "systematic differences in health between social groups, that are avoidable by reasonable means" [13], often found on both the patient level as well as on that of health professionals and organizations (i.e. the healthcare system and social welfare system) [14].

Patients may have trouble interpreting and reacting adequately on their symptoms and adverse events, and they may lack trust in the healthcare system, which results in non-attendance or non-compliance [14–16].

Lack of screening for and management of mental illness before initiating cancer treatment constitutes a barrier on the level of health professionals. Additionally, insufficient knowledge and competencies of oncology staff regarding mental illness further exacerbate this issue [14–16]. Stigmatization is also an important barrier in this regard, either constituted by visible differences in behaviour towards the stigmatized person or more subtle actions by well-meaning people who are unaware of the potentially harmful effects. Stigmatization can potentially result in delayed diagnosis and treatment among the patient group [17].

On the organizational level, the barriers are mainly a fragmented healthcare system, the lack of a systematic approach to the patient group, and lack of resources and continuity of care [14–16].

Although relevant, the barriers described above are largely found without investigating the patients' experiences. With the ongoing emphasis on the importance of shared decision making where patient and clinician partner up in decisions on treatment and care [18] the patient perspective becomes increasingly important also in research. Research has already shown that shared decision making has the potential to improve the quality of care to patients with severe mental illness [19, 20]. Practising shared decision making is moving healthcare from giving everyone the same treatments to patient-centred care where treatment and care is individually tailored to patients' health needs and includes their perspective in treatment decisions [21]. Simply attempting to ensure that all patients receive guideline-concordant treatment might be an outdated goal. Equity to a larger degree might be achieved by providing patient-centred care for patients with mental illness and cancer.

In this study, we investigated the barriers and facilitators to patient-centred cancer treatment and care for patients with simultaneous severe mental illness.

## Methods

Inspired by a hermeneutical approach as presented by Hans-Georg Gadamer [22, 23], this qualitative case study explores the lived experiences of cancer treatment and care in patients with severe mental illness. The study aims to shed light on barriers and facilitators in providing patient-centred care in daily clinical practice. Congruent with the traditions of hermeneutics, our pre-understanding are prerequisites for understanding the phenomena of mental illness, cancer, and barriers and facilitators in providing patient-centred care, and thereby we are co-producers of the knowledge provided in this study. Accordingly, our pre-understandings have played an important role in the planning and execution of this study, but as highlighted by Gadamer, new understandings that integrates elements from both patients' and our own understandings have also emerged. Being conscious and transparent about our pre-understanding is crucial for a deeper understanding [23]. Therefore, first author (AV) discussed her pre-understanding and reflected on its impact on the data with author MS in the initial phase of the study. AV, a medical doctor with experience in oncology, psychiatry and general practice, brings a unique perspective to the study. Her previous work with vulnerable patients from ethnic minorities and patients with mental illness, respectively, has given a pre-understanding of the healthcare system as being poorly equipped to care for patients with multiple challenges. Her pre-study concerns included limited cross-sectorial coordination, patients' irregular adherence to appointments and instructions and their lack of trust towards the staff, as well as health professionals' challenges due to lack of knowledge and competencies in caring for patients with mental illness.

Focusing on functional impairment rather than specific diagnoses, we chose to follow the definition of severe mental illness by the National Institute of Mental Health: "A mental, behavioural, or emotional disorder resulting in serious functional impairment, which substantially interferes with or limits one or more major life activities" [24]. In line with this definition, participants in this study were recruited from secondary psychiatric services, including both inpatient and outpatient settings, with the additional criterion that they had a history of prolonged and complex engagement with secondary psychiatric care.

We intended to investigate the patients' experiences particularly related to being in the field between oncology, psychiatry, general practice, and municipality. Field observations, semi-structured qualitative research interviews, and patient files were used as data. The first author collected all the data.

All authors of this paper are experienced health professionals, and all but two authors worked at the investigated departments (Oncology and Psychiatry) during the study.

Findings and important considerations during data collection were discussed with a patient representative.

The study is reported according to the Equator networks "Standards for reporting qualitative research: a synthesis of recommendations" [25].

## Study setting

Recruitment and data collection took place between December 19th 2022 and June 27th 2023. The study was conducted in Denmark at hospitals and municipalities in the Region of Southern Denmark.

In Denmark, Danish citizens receive free healthcare and social security, funded through taxes.

Patient files are managed electronically but in separate systems in the hospital, general practice, and municipality settings. Information is typically exchanged between the hospital and general practice through written correspondence or summarized in a discharge letter. In the

Danish healthcare system, the general practitioner (GP) plays a central role for each patient, serving as the primary care provider, managing overall health, and coordinating care across sectors.

The psychiatric system in Denmark consists of inpatient and outpatient clinics. Care-taking staff often includes nurses, social educators and social and health assistants.

The Department of Oncology, Vejle Hospital, was the main area for observations. It comprises two inpatient clinics, several outpatient clinics for medical treatment and radiotherapy and offers integrated specialized palliative care services.

The municipalities are responsible for home-based care services, social psychiatry, social security, occupation and child services. The specific services in each municipality vary considerably. At Vejle municipality, four social nurses work specifically with vulnerable people with mental illness. Their job includes screening for somatic diseases, following the patients to appointments in the healthcare system, and coordinating their treatment trajectories. The aim is to help the patients through treatment and facilitate a closer connection with the GP and the healthcare system in general.

## Participants

Five patients were recruited through purposeful sampling, representing variation in gender, age, oncologic and psychiatric diagnoses and prognosis (Table 1). All invited patients accepted and provided data to the study. Recruitment concluded when nuanced descriptions and perspectives were obtained, and recognizable patterns emerged, thus reaching data saturation.

During the study, three patients were undergoing curative or adjuvant oncology treatment, while one patient was receiving life-prolonging medical oncology treatment. Another patient, who had been discontinued from adjuvant oncology treatment, was instead being monitored with control CT scans at the time of their inclusion in the study. This patient was still considered able to provide valuable information regarding the treatment trajectory and was therefore retained in the study.

Three patients were undergoing radiotherapy, and three patients (with two overlapping with radiotherapy) were receiving medical oncology treatments, administered either as injections or as oral tablets. Additionally, three patients had previously undergone surgical removal of their cancer, and two had received intravenous chemotherapy before the study began.

Two patients were undergoing electroconvulsive treatment (ECT), and all patients were receiving psychopharmacological treatment.

**Table 1. Characteristics of patient participants in the study.**

| Patient identifier | P01 | P02 | P03 | P04 | P05 |
|---|---|---|---|---|---|
| Sex (Male/Female) | M | F | F | F | M |
| Age group (yrs) | 50–59 | 40–49 | 70–79 | 50–59 | 50–59 |
| Psychiatric diagnoses | PTSD Depression | Depression Anxiety | Bipolar disorder | Personality disorder Anxiety Reaction to severe stress | Schizophrenia Bipolar disorder |
| Cancer type | Prostate | Breast | Ovarian | Endometrial | Lung |
| Relative(s) present during field observations? | No | Yes* | Yes | No | Yes |
| Relative(s) participating in interviews? | No | No | Yes | No | No |
| Time spent observing (hours) | 12 | 10 | 7 | 2 | 3 |
| Time spent on interviews (min) | 54+ | 40 + 4 | 19 + 29 | 43+ | 46+ |

*Voluntary support person

## Data collection

**Field study.** Spradley's framework for participant observation was used for the field study [26]. AV followed the included patients during visits at the Department of Oncology, Vejle Hospital, and at psychiatric departments in the Region of Southern Denmark. Two patients were additionally followed during visits at the Department of Radiology, Vejle Hospital for CT and MRI scans. We intended to follow the patients to their GP and the municipality, but the patients either had no appointments there, or did not wish to be observed in those settings.

Since it proved difficult to follow the patients in more than two settings (psychiatry and oncology), we arranged meetings with, and observations of, professionals with knowledge of cross-sectorial work and the patient group at the two departments and Vejle municipality. This included doctors and nurses at the department of oncology and psychiatry, discharge- and treatment trajectory coordinators at the departments, and social nurses at the departments and in the municipality. The observations lasted between 30 minutes and 8 hours.

In total, AV did field observations with patients for around 35 hours and the staff around 25 hours.

Following Spradley's approach [26], before initiating field observations with the patients, AV gained an initial overview of the social situation by carrying out grand tour observations of nurses working at one of the inpatient clinics at the Department of Oncology. These observations gradually gave an idea of specific aspects to investigate in more detail through mini-tour observations, e.g. how the mental illness was addressed in oncology consultations and interactions between health professionals and patients during procedures.

During observations, condensed accounts were written and later elaborated in expanded accounts either immediately after or within two days, describing the observations in as much detail as possible. A logbook of thoughts and ideas was kept as an account of initial analyses.

AV did not wear a uniform during field observations in order to signal that she did not represent health professionals on duty but was there as an external observer.

To answer the research questions and supplement information from patient interviews and observations, the following information was retrieved from the participants' patient files:

1. In the cancer trajectory, is the oncology staff aware of the psychiatric diagnosis and is it visibly considered in the treatment plan?

   a. Is the psychiatric diagnosis mentioned in the referrals, visitation notes, first consultation at the department of oncology and in the multidisciplinary conference-notes?

   b. In the types of notes described above, is it mentioned whether the patient is undergoing psychiatric treatment or otherwise connected to the psychiatric setting?

   c. In the types of notes described above, is it mentioned whether the patient is receiving psychopharmacologic treatment?

   d. If any of the above are mentioned in the notes, is it visibly addressed whether this impacts the oncologic treatment plan?

2. How is the role of the GP described in correspondences and discharge letters?

This approach was chosen in order to have a clear focus when going through the patient files and avoid reading more of the patient files than necessary, preserving as much of the patients' privacy as possible.

**Interviews.** The overall aim of the interviews was to allow patients to discuss and elaborate on their thoughts and experiences. Observations and findings from the field observations were included and discussed in the interviews.

Following Kvale & Brinkmann [27] a semi-structured interview guide was developed in advance by the first and second author (S1 Table). After the first interview, small adjustments were made in terms of wording of some of the questions.

All patients were interviewed at Vejle Hospital before or after their appointments according to their wishes. We planned to interview each patient once, but in two cases, a follow-up interview was carried out because the initial interview ended prematurely. Interviews lasted between 20 and 60 minutes and were performed in a private room connected to the oncology department. The interviews were tape-recorded and transcribed verbatim.

A translator participated in one of the interviews, since Danish was not the patient's first language.

## Data analysis

Following Spradley and Kvale & Brinkmann [26, 27], a thematic analysis was carried out focused on meaning. Data collection and analysis were conducted concurrently, allowing findings from ongoing interviews and field observations to inform each other.

Field notes, interview transcripts and patient file excerpts were initially read and listened through for a sense of the whole picture and coded following the approach by Spradley [26]. Based on the codes, themes were developed and explored further by comparing them across the data.

Open coding was used with no codes developed before examining the data. AV was in charge of coding and analysis of the data and frequently discussed codes and findings with the second author (MS). The rest of the team, including the patient representative, was also presented with emerging codes and themes and provided feedback. All data was coded using the qualitative data management software NVivo R14.23.2.

## Ethics

The storing and handling of data was approved by the Region of Southern Denmark (jr. nr. 21/67977). According to the Regional Committee on Health Research Ethics for Southern Denmark, their approval of the study was not required (ref. 20212000–184) since it does not involve an intervention in the sense of the Committee's Act.

Patients were enrolled on the basis of written and orally informed consent. Personal data were stored and managed in compliance with Danish legislation and GDPR through the technical facilities of OPEN (Open Patient data Explorative Network), Odense University Hospital, Region of Southern Denmark [28]. Patient files were accessed during data collection (December 19th 2022 to June 27th 2023) after having obtained written and oral consent from participants. Pseudonymized excerpts were transferred to NVivo where only the authors had access to the data. Only researchers involved in the project had access to the data both during and after data collection, including identifiable information.

Patients could decide the observer's presence and cancel interactions without affecting their care. Permission from the patient was sought again during changes of scene. Uniforms were avoided to reduce power imbalance, and discussion of personal topics was not initiated in public areas.

## Results

Five patients were included and the characteristics are given in Table 1. The study revealed that defining separate barriers in a cancer trajectory for the patient group was very complicated, as many of the discovered barriers were woven together and spanning across the levels of patient, health professional and health care system. This was illustrated in the main theme

"Complexity on many levels" with four subthemes describing important areas for barriers and facilitators:

- How the mental illness is affected by the cancer trajectory

- The complexity of patient vulnerability

- Fragmented healthcare system and lack of structure

- The role of the relationship between patient and health professional

## Main theme: Complexity on many levels

Complexity and sometimes paradoxical issues seemed to be present in almost every situation in the patients' trajectories: The patients' reaction to and perception of the cancer treatment, the nature of their own resources and limitations, the healthcare system's way of handling the patients, and the influence on the cancer trajectory by the relationship between patients and the health professionals.

Even the patients themselves were verbal about the complexity. When asked what they would have wished differently in their trajectory, they all responded that this is a very complex issue, that no one could have known that they would face these challenges, and that finding a solution was not a simple task. The following describes the four subthemes in order to shed light on important barriers and facilitators and their complex connections.

**Subtheme 1: How the mental illness is affected by the cancer trajectory.**   An important theme was that the cancer diagnosis or treatment trajectory sometimes seemed to worsen the psychiatric symptoms to a degree that led to acute hospitalization at the Department of Psychiatry. This was described both by the patients and health professionals, written explicitly in the patient files and in one instance observed during field observation. In some instances it resulted in altered cancer treatment.

Different parts of the cancer trajectory served as possible triggers for worsened mental symptoms, e.g. a diagnosis of non-curable cancer recurrence, as well as being overwhelmed by side effects of the treatment or the haste of diagnosis and treatment. The triggers were often interwoven, difficult to distinguish, and dependent on issues present before the cancer diagnosis. Very often the patients' worry and stress of trying to determine whether what the patients were experiencing were normal, manageable side effects or a sign of something alarming they needed to react to, was a large trigger in itself.

The patients often expressed that the health professionals were unable to help them determine this even after the patients had voiced their concerns, leaving them feeling even more helpless:

*"I'm thinking, Could it be my mind? Or could it be something physical? Could it be. . . (Pause) Residual chemo, could it be antibodies, could it be radiation, could it be something else entirely? I mean, it just goes on like that constantly, and no one can give me any answers"* (P02)

Referring the patients for psycho-social support early on was suggested by patients as a possible solution as well as helping them meet other patients with cancer in order to find out whether they had similar experiences.

Paradoxically, the patients also clearly expressed that the cancer diagnosis was inferior to the psychiatric diagnosis and context:

*Field notes, patient P01:*

P01:"I don't think about the cancer at all. [. . .] As long as I don't have any pain, I really don't care. [. . .] As long as I'm well mentally it's fine."

Interviewer: "What then when you are not well mentally?"

*P01*: *"Then I want to die."*

For some patients the threat of the mental illness was constituted by the fear of its treatment. Especially the fear of not waking up after ECT was expressed as taking up more mental capacity than the cancer trajectory:

*"In a way it's the other thing (ECT) that takes up more space (than the cancer trajectory, red.). [. . .] Because I'm so afraid of it." (P03)*

This did not, however, make the patients take the cancer diagnosis less seriously. The patients clearly described the cancer treatment as very important and were observed to be very adamant on showing up for the appointments even if it took them days of reminding themselves. The patients' relatives were a big factor in this as will be presented in subtheme 2.

It seems that the anxiety connected to both the mental illness and the cancer revolves around the fear of death, whether it is by suicide, not waking up from ECT or dying of cancer. Most of the patients expressed worry about the risk of dying of cancer, especially if they were considering discontinuing treatment:

*Field notes P02:*

[P02] says she had considered declining the offer of radiotherapy because she just can't take anymore, but that she is also afraid that if she declines she will constantly have the fear "Has the cancer come back?"

Interview P03:

Interviewer:"Does anything about the cancer worry you? I mean, do you think about it?"

P03: "Yes, I am nervous about it, all the time—of course. I mean, about whether there is something there."

*Relative*: *"Well, it comes up a lot. I mean. . . If there is pain somewhere. . . 'Is that the cancer?'"*

Even when they spoke of being exhausted or after having been hospitalized at the Department of Psychiatry, it was the oncology health professionals who suggested the cancer treatment be discontinued, not the patients.

How one is going to die obviously matters a great deal, but whether one threat is worse than the other seems to vary depending on how often one is confronted with it. For instance, having weekly ECT appointments may cause a patient to fear dying every week as opposed to a cancer that is presumably under control while being treated. The psychiatric illness seems to be a known and very distressing way of passing away, whereas dying of cancer is an unknown, more abstract way that in some ways is out of the patient's control. The cancer treatment was often expressed as being managed by the health professionals, whereas it seems the mental illness is something the patients are faced with and need to deal with on a daily basis. For some, the fear is not so much of death itself but rather the fear of leaving children and relatives behind.

**Subtheme 2: The complexity of patient vulnerability.** The nature of the patients' vulnerability was observed to vary considerably from day to day in our study. Sometimes asking for

help became extremely difficult for the patients, requiring others to intervene. On other occasions, the patients were able to respectfully and continuously voice their needs to the health professionals, which was greatly appreciated by the health professionals during field observations.

The patients rarely seemed to see themselves as vulnerable. In interviews and during field observations, they often described having used large amounts of resources on raising children and holding down full-time jobs, some even managing to get into and complete demanding educations, while not allowing the mental illness to affect them.

*Field notes P01:*

*P01:"People have asked me:"How can you go through all that (the mental illness trajectory, red.) and still have children and work etc.?". He pauses and explains that he has pushed it all down. But that takes energy. And in the end he couldn't any longer."People have offered many times (to take him to the psychiatric emergency department, red.). But I didn't want to do it before because of my children."*

They all described a point of reaching the limit of their capacity, resulting in a lack of resources that presented itself by being easily stressed by matters such as information overload, sitting in large spaces with other people and keeping track of multiple appointments in different departments. Patients would often spontaneously describe concrete issues related to their mental illness but would then omit or understate them when asked during interviews if they faced any challenges during the cancer trajectory. It became clear that it mattered to the patients being viewed not as vulnerable victims but as normal human beings facing specific challenges. This was also underlined by the social nurses at the municipality:

*Field notes of social nurse at municipality:*

*Social nurse:"What the citizens need the most is not to be viewed as psychiatric patients but to be seen as people with specific problems that need to be solved"*

The relatives and loved ones served as a large part of the patients' resources; sometimes specifically as necessary coordinators in the treatment trajectory, sometimes as the main reason to carry on and continue cancer treatment. Some patients expressed feeling guilty about relying too much on the relatives and indicated wanting to do almost anything to avoid burdening them:

*Field notes, P01:*

Interviewer: "It sounds like your children are the most important thing to you; that you feel your children are more important to you than yourself."

P01 nods and says that [. . .] each of them means as much to him as himself. He has experienced feeling really awful and really wants his children to avoid that.

*P01: "While they are small you have a responsibility to show them life. I have struggled really hard. And I still do."*

**Subtheme 3: Fragmented healthcare system and lack of structure.** The healthcare system had issues of fragmentation and lack of structure, particularly in terms of lack of coordination and collaboration between departments and the GP, different ideas on what the staff was

allowed to access in patient files, and an unclear role of the psychiatric diagnosis in the oncology treatment plan.

The health care system had limited coordination of treatment trajectories and collaboration between the psychiatric and oncology departments and between the departments and the GP. The oncology staff described that communication with the psychiatric staff was experienced as difficult because they rarely understood the terms and procedures used in a somatic setting:

*Field notes treatment trajectory coordinators, Department of Oncology:*

*"You sometimes really feel that we are two different worlds. You have this expectation that when you're talking to a nurse we can speak at a certain technical level, but then you sense on them that some of what you are saying is practically Russian"*

The staff at the psychiatric department expressed that somatic departments generally show little understanding for the fact that psychiatric staff have other important skills that differ from the ones required in a somatic setting, and sometimes they feel they are not taken seriously when they need somatic assistance, e.g. when needing to refer the patients for diagnosis of a possible cancer recurrence.

*Field notes, nurses at a psychiatric department:*

*[The nurse] explains that they have had patients with a history of cancer [. . .] where they have had to very quickly rule out a cancer recurrence in order to proceed with psychiatric treatment. [. . .] In those cases they have had to wait several days before even hearing from the somatic departments. That during the waiting time the patients can experience such a severe psychiatric worsening that they eventually have to treat the patients with coercive ECT.*

An important issue was uncovered in terms of access to patient files outside one's own department. Some psychiatric staff stated that they were not allowed to access somatic patient files, except on rare occasions with direct relevance to the psychiatric treatment. They were then required to write explicitly in the file why they had gained access. The staff presented this as a legal requirement but then explained its importance in terms of upholding trust and protecting the patient and staff, thus emphasizing it as a moral obligation. This conception was not shared by the oncology staff. We found no official documents describing this rule. Even when contacting the legal department of the Region of Southern Denmark, we only received official documents describing how it is always a professional decision whether reading patient files from external departments is relevant to the current treatment and care. This implies that part of the fragmentation is a cultural issue rather than a legal one, but the misconception could potentially worsen the lack of collaboration and coordination between departments.

The lack of structure was especially observed in the oncology field regarding how to handle the knowledge of the mental illness. In patient files, the health professionals often showed awareness of the psychiatric diagnosis and/or current psychiatric treatment status, but rarely wrote explicitly whether the psychiatric diagnosis and treatment should have any consequences for the cancer treatment.

When asked whether the oncology staff would want themselves or their colleagues to be aware of the mental illness, they often expressed a worry that disclosing the psychiatric diagnosis could negatively affect the patient in the form of discrimination and stigmatization and that they needed to determine what would harm the patient the least.

*Field notes, treatment trajectory coordinators, Department of Oncology:*

*[The coordinator] says that [. . .] "sometimes it might be better to meet the patient where they are". That she can sense that with a patient with borderline personality disorder, they currently have at the department, the conversation changes because she (the coordinator) knows about the diagnosis.*

This however, does not seem to explain the reason for disclosing the psychiatric diagnosis in the treatment files but not the consequence for the treatment plan.

During field observations, the lack of interdepartmental collaboration and structure often resulted in insufficient treatment coordination. With everyone responsible and no official coordinator, individual health professionals were often left to deal with arising issues during busy everyday procedures with very limited information about the patient and situation to work from. This often resulted in extra work for both departments but no proper management of the issue. In some cases it seemed that better structure and collaboration could have prevented some of the psychiatric hospitalisations described in subtheme 1.

The general practitioner generally had an undefined role in the trajectory. According to excerpts of patient files, the GP was expected to follow up on issues that had some connection to the treatment at the hospital department (both psychiatry and oncology) but was rarely involved before discharge. During interviews and field observations, the patients expressed how they had been told explicitly by the health professionals that the GP is for matters outside psychiatry or oncology and that they were discouraged to talk about issues in the "wrong setting". This ruled out the GP as the natural coordinator in the treatment trajectories.

*"Now that [P03] has this back pain. . . They tell us immediately 'That's the GP'. And the same with the GP who says '[P03], let's not meddle with that' I mean. . . 'They know about that at the psychiatric department'" (Relative of P03)*

The patients were observed to deal with this fragmentation by simply discussing all their medical issues with the staff they trusted and felt had the responsibility for the somatic treatment, mainly the oncology staff. This seemed to be an extension of the need to be viewed as a person rather than separate diagnoses:

*Field notes, P01:*

*P01: "He (the doctor) says that they (the issues) are not connected. As if they belong to two different people. But they do not belong to two different people. Both issues are within me."*

**Subtheme 4: The role of the relationship between patient and health professional.**
Trust, mistrust and stigmatization were important issues in the treatment trajectory. Several of the patients described experiences of being stigmatized by either doctors or the municipality staff. Some expressed that their diagnosis or treatment was delayed because their somatic symptoms were assumed to be due to their mental illness. This often impacted their view on the respective institution in general, resulting in the patients deliberately avoiding seeking help with for instance the GP or municipality regardless of whether the staff was the same or had changed. One patient gave this simple explanation:

*"If you don't want a rap by a stick, you don't run over to the man holding it." (P05)*

The social nurses at Vejle Municipality emphasized trust as a key ingredient in their work with the patient group. They often deliberately chose to first deal with the issues most

important to the patient and set aside what the nurses considered the most pressing issue. The trust gained from this often made the patients more likely to be motivated for then handling the health concerns that the nurses found most important.

During field observations, the hospital staff was observed to treat the patients with great attention and patient-centeredness despite the issue of the fragmented healthcare system. The nurses moved between assessing symptoms, performing procedures, and asking and listening with presence to the patient's concern. During consultations doctors and nurses at both types of departments used open body language signalling attentive listening. When asked about their thoughts on their approach to the patient, the staff often responded that the patient is "such a sympathetic person" and that it was "unbearable to see them suffer", or that they could recognize what the patient was feeling or going through from their own lives.

It seems that the majority of the observed staff first and foremost viewed and treated the patients as humans; something that was felt by the patients. Patients expressed spontaneously in interviews and field observations how the staff seemed "so nice" and like "good people" and how much the patients appreciated being cared for by staff they knew well and trusted:

> *"I have felt incredibly well-treated here [at the Department of Oncology]; I really have. Such kind people, all of them, and. . . And I think that makes such a big difference, you know. . . That you are not just treated as a patient. . . (Pause) But as a human being." (P02)*

### Barriers and facilitators summarized

The barriers and facilitators presented in this study are summarized in Table 2:

## Discussion

This study elucidated significant barriers and facilitators in cancer trajectories for patients with severe mental illness, providing valuable insights for enhancing patient-centred care.

**Table 2. Barriers and facilitators for patients with severe mental illness in a cancer trajectory uncovered in this study.**

| Barriers | Facilitators |
|---|---|
| Interwoven patient problems | Using peers to help the patients realize which problems might normally arise in a cancer trajectory and which might be due to their mental illness |
| Exacerbation of psychiatric symptoms by cancer treatment | Consistency in healthcare staff |
| Difficulty in identifying a single source of responsibility | Uncovering the patients' resources and determining how to handle any day to day variation in the patient's vulnerability |
| Poor communication and coordination between oncology and psychiatry | Understanding the role and importance of the patients' relatives |
| Lack of general coordination in the trajectories | Building of trust and a personal connection with patients |
| Lack of structure for dealing with the patient group, resulting in changing, individual staff having to handle the issues | Social support and relevant care from the municipality |
| General practitioner not effectively integrated into the care continuum | |
| Depletion of patient resources due to managing multiple conditions | |
| Patient mistrust towards the municipality and GP | |
| Stigmatization of psychiatric patients | |

Uncovered barriers included fragmentation of the healthcare system, a lack of a systematic approach to the patient group, and severe worsening of the mental illness during the cancer trajectory. Facilitators included the importance of the health professionals acknowledging the patient's own resources and approaching the patient as a person rather than a disease.

Our study is one of the few to date investigating the patient perspective in cancer treatment and care for patients with severe mental illness. Previous studies have mainly described barriers and facilitators for the patient group as perceived by health professionals or focused on cancer screening [14–16]. One fairly new qualitative study has investigated the patient perspective specifically [29]. In line with our findings, the studies have found that the cancer diagnosis and treatment for some patients can involve significant deterioration in their mental health [15, 29]; our study went further by uncovering specific triggers for the deterioration, thus providing valuable knowledge for developing guidelines on planning and monitoring cancer treatment for the patient group. The findings suggest a need to learn more about the impact of the cancer trajectory on mental illness and educating health professionals on how to integrate knowledge of the mental illness in the treatment plan, mirroring the structured approach to somatic comorbidities such as impaired renal function, and their impact on the cancer treatment plan.

Our study's findings are in line with findings by Leahy et al. (2023) [29], revealing complex patient vulnerability consisting of a combination of resilience, mental strength and struggles with voicing their needs. This nuanced view of patient resources and vulnerability could be important in terms of ending stigma and increasing patient involvement in the patient group instead of primarily describing the patients' vulnerability in the form of issues such as health literacy and non-compliance.

Fragmentation in the healthcare system with lack of coordination, resources and structure for dealing with the group is a known and well-described theme in the literature [14–16]. However, in line with Leahy et al. (2023) [29] our study went further and examined the problem and possible facilitators as perceived by the patients. We learned how patients bypassed the fragmentation in spite of specific instructions to separate their issues and still suffered when important issues were not handled by any of the involved departments. Our study revealed specific structural and cultural target areas that will likely ease the implementation of changes to enhance collaboration and coordination. This could include deciding on resource allocation and structure for dealing with the patient group within the departments and securing consistency in healthcare staff. On a larger scale, a main coordinator of the trajectories could be appointed, and a way to increase the collaboration between the departments, general practice, and the municipality could be planned.

Our study showed not only the perceived importance of approaching the patients as humans rather than diagnoses but also the impact on the patients. Shared decision making with patients is already quite prevalent in the Region of Southern Demark, and other person-centred approaches such as "Excellent nursing" are actively taught at the Department of Oncology. "Excellent nursing" is an approach where awareness of patients' experiences and needs is balanced with biomedical observations, and includes a training program involving reflections on one's own practice. This might be one reason for the positive relationships between patients and health professionals and the observed approach of seeing the whole person. The departments, however, still faced multiple barriers in the patient trajectories, suggesting that this patient-centred approach in itself is not enough to deal with the inequity experienced by the patient group. Rather, solutions taking several levels of barriers into account might be needed in order to properly solve the issue.

Such a solution has been investigated in a recent clinical trial by Irwin et al. [30] examining the impact of a patient coordinator and cross-sectorial collaboration in the BRIDGE trial.

Their work includes early detection of mental illness and assessment of patient and caregiver strengths and goals for treatment. Preliminary results have shown significantly fewer disruptions in cancer care and reduced severity of mental illness [31].

A combination of structure and patient-centeredness has been developed by Weiner and Schwartz and termed "Contextualising healthcare" [32]. In this approach, doctors are trained in detecting and addressing barriers towards access to healthcare that arise from the patient's context, e.g. financial issues or troubled relationships with healthcare professionals. In our patient group, the doctor would focus less on the psychiatric diagnosis and more on the specific issues that were barriers to treatment [32]. Given the complexity of the barriers and how the issues were interlinked with the patients' comorbidities and social factors, this approach might prove applicable in helping the staff help the patients.

## Strengths and limitations

A strength of the study is the detailed look at the patient's trajectory in the form of data triangulation–including both formal and informal interviews with the patients, field observations, patient file analysis, and informal interviews and observations of health professionals with relation to the patient group. This provided information on the personal, structural and cultural barriers in the trajectories that would not have been revealed had only one modality been chosen. This in-depth work required a low number of participants, but enrolment of patients was only stopped when nuanced descriptions and perspectives were obtained and data saturation seemed to have been reached. The themes that emerged seemed to span across data from all patients regardless of diagnoses and demographics; the themes reflected barriers that in some ways are somewhat fundamental to all patients–such as fragmentation of the health care system and the importance of the relationship between the health professionals and patients–but were of particular importance for this patient group, as shown by the perceived consequences of some of the barriers; lastly, the uncovered themes were in line with–and seemed to expand on findings from–literature in the field–all of which speaks to a high degree of transferability.

In order to enrol patients with different challenges and resources, we chose to let the patients decide for how long and in what settings they were observed. As all invited patients accepted and did not withdraw from the study, and a large difference in observation time was seen (Table 1), we believe that this has allowed a wider spectrum of patients to be represented in our study than would otherwise have been possible.

The frameworks used for the study (Spradley [26] and Kvale & Brinkmann [27]) are widely acknowledged and were chosen–along with the hermeneutic approach–in order to best answer the research question. Findings were discussed both among the authors and with the patient representative in order to further ensure internal validity.

The geographic location of the study could mean that some results would mainly concern the Danish Healthcare system, but since our findings were in line with recent international literature this did not seem to be an issue.

AV worked one year as a medical doctor at the Department of Oncology before starting the project and was familiar with many of the observed health professionals at the department. This seemed to allow her to ask staff members more freely about their thoughts and experiences and allowed her to listen in on their conversations during procedures. The connection might have influenced how highly the patients spoke of the department, but the patients readily pointed out flaws and problems at the department without seeming to worry about AV's connection. On occasion, the health professionals (mainly nurses) asked about AV's opinion on a matter because they knew she was a doctor, but respected when AV explained her role of being exclusively an observer.

In one instance, an observed doctor later explained that she had avoided talking about the patients' mental health concerns because she thought AV was a support-person in that regard.

AV's role as a doctor, and thus previously larger degree of responsibility, occasionally resulted in a moderate involvement, reminding the patients of things they had forgotten to ask about or encouraging them to check if they were waiting in the right area. Fortunately, the interfering seemingly had little impact on the situation, as the questions were minor and the patients often had already understood things correctly and needed no guidance.

We chose not to investigate patients with ADHD or dementia, as they do not have prolonged and complex engagement with the secondary psychiatric care. We acknowledge that these patient groups still face issues during cancer treatment, but we anticipated their experiences to differ too much from the other patient groups.

## Conclusion

This study underscores the complexity of care for patients with cancer who also suffer from severe mental illness, revealing key areas for improvement. Our analysis identified the overarching theme of "Complexity on many levels," with four subthemes: the impact of the cancer trajectory on mental illness, the multifaceted vulnerability of patients, the fragmentation and lack of structure in the healthcare system, and the critical role of the patient-health professional relationship. Barriers to effective care included the exacerbation of mental illness due to the cancer trajectory and systemic fragmentation. Facilitators included recognizing and leveraging the patient's own resources and adopting a person-centered approach to care. By addressing these barriers and enhancing these facilitators, healthcare providers can better meet the needs of this vulnerable population.

## Supporting information

**S1 Table. Interview guide.**
(DOCX)

## Acknowledgments

We would like to thank patient representative Henrik Frost Hansen for invaluable feedback and reflections on both research approach and analysis.

This work used the technical facilities of OPEN (Open Patient data Explorative Network), Odense University Hospital, Region of Southern Denmark [26].

## Author Contributions

**Conceptualization:** Astrid Næraa Høeg Vendelsøe, Mette Stie, Peter Hjorth, Jens Søndergaard, Dorte Gilså Hansen, Lars Henrik Jensen.

**Data curation:** Astrid Næraa Høeg Vendelsøe.

**Formal analysis:** Astrid Næraa Høeg Vendelsøe, Mette Stie.

**Funding acquisition:** Astrid Næraa Høeg Vendelsøe, Mette Stie, Peter Hjorth, Jens Søndergaard, Dorte Gilså Hansen, Lars Henrik Jensen.

**Investigation:** Astrid Næraa Høeg Vendelsøe.

**Methodology:** Astrid Næraa Høeg Vendelsøe, Mette Stie, Dorte Gilså Hansen.

**Project administration:** Astrid Næraa Høeg Vendelsøe, Dorte Gilså Hansen, Lars Henrik Jensen.

**Resources:** Astrid Næraa Høeg Vendelsøe, Lars Henrik Jensen.

**Supervision:** Mette Stie, Peter Hjorth, Jens Søndergaard, Dorte Gilså Hansen, Lars Henrik Jensen.

**Validation:** Mette Stie.

**Visualization:** Astrid Næraa Høeg Vendelsøe.

**Writing – original draft:** Astrid Næraa Høeg Vendelsøe.

**Writing – review & editing:** Astrid Næraa Høeg Vendelsøe, Mette Stie, Peter Hjorth, Jens Søndergaard, Dorte Gilså Hansen, Lars Henrik Jensen.

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
