## [Decision Letter · Decision Letter 0]

18 Jul 2024

PONE-D-24-23981”I just felt that everything came tumbling down around me” - Barriers in cancer care for patients with severe mental illness: A qualitative studyPLOS ONE

Dear Dr. Jensen,

Thank you for submitting your manuscript to PLOS ONE. After careful consideration, we feel that it has merit but does not fully meet PLOS ONE’s publication criteria as it currently stands. Therefore, we invite you to submit a revised version of the manuscript that addresses the points raised during the review process.

**ACADEMIC EDITOR: **This manuscript has merit and would be of great interest to readers. There are few areas that require further clarification before it can be considered for acceptance:Reviewer 2 has queried regarding the 'hermeneutical approach' employed in this manuscript. It would be advisable to explicitly state the characteristics of hermeneutical approach in the methods. I do note that the manuscript has reported the application of reflexivity and interpretive analysis. Rephrasing the methods to be more explicit regarding the hermeneutical approach may be helpful.The definition of severe mental illness was provided in lines 97-99. As mentioned by Reviewer 1, the application of this definition of severe mental illness would require clarification. The participants of the study consisted of patients with mood disorder, anxiety disorder and one patient with psychotic disorder. All these could have a range of mild to severe illness, as patients with severe mood disorder may also have psychotic symptoms. Elaboration of the severity of these diagnoses may help to justify the definition for severe mental illness. Additional quotes to support the interpretation of results as mentioned by Reviewer 1 is required.In line with hermeneutical approach, providing more context would be helpful. For example, which patients had undergone ECT? What cancer phase were the participants in at the time of the study? Had they undergone surgery, chemotherapy, or radiotherapy? Why did the study specifically include the role of GPs as it was not evident in the objectives?==============================Please submit your revised manuscript by Sep 01 2024 11:59PM. If you will need more time than this to complete your revisions, please reply to this message or contact the journal office at plosone@plos.org. Please include the following items when submitting your revised manuscript:A rebuttal letter that responds to each point raised by the academic editor and reviewer(s). You should upload this letter as a separate file labeled 'Response to Reviewers'.A marked-up copy of your manuscript that highlights changes made to the original version. You should upload this as a separate file labeled 'Revised Manuscript with Track Changes'.An unmarked version of your revised paper without tracked changes. You should upload this as a separate file labeled 'Manuscript'.

We look forward to receiving your revised manuscript.

Kind regards,

Chai-Eng Tan

Academic Editor

PLOS ONE

2. PLOS requires an ORCID iD for the corresponding author in Editorial Manager on papers submitted after December 6th, 2016. Please ensure that you have an ORCID iD and that it is validated in Editorial Manager. To do this, go to ‘Update my Information’ (in the upper left-hand corner of the main menu), and click on the Fetch/Validate link next to the ORCID field. This will take you to the ORCID site and allow you to create a new iD or authenticate a pre-existing iD in Editorial Manager. Please see the following video for instructions on linking an ORCID iD to your Editorial Manager account: https://www.youtube.com/watch?v=_xcclfuvtxQ".

3. We notice that your supplementary table are included in the manuscript file. Please remove them and upload them with the file type 'Supporting Information'. Please ensure that each Supporting Information file has a legend listed in the manuscript after the references list.

Reviewers' comments:

Reviewer's Responses to Questions

**Comments to the Author**

1. Is the manuscript technically sound, and do the data support the conclusions?

Reviewer #1: Yes

Reviewer #2: No

2. Has the statistical analysis been performed appropriately and rigorously? 

Reviewer #1: Yes

Reviewer #2: No

3. Have the authors made all data underlying the findings in their manuscript fully available?

Reviewer #1: Yes

Reviewer #2: No

4. Is the manuscript presented in an intelligible fashion and written in standard English?

Reviewer #1: Yes

Reviewer #2: No

5. Review Comments to the Author

Reviewer #1: Thank you very much for the opportunity to review this article. Overall, it is a very good piece and it is great to see more work is being done in this area. I have a few notes (please see below).

1. Introduction: you mention both mental illness and severe mental illness (SMI). SMI has been referred to as schizophrenia and bipolar, however, no examples have been given for mental illness – should all of this say severe mental illness? Also, more background could be given on what severe mental illness is. Such as what is schizophrenia/bipolar etc and how many people are affected by it? Are there any other severe mental illnesses?

2. Methods: It is not clear if the patients were diagnosed with a SMI or a mental illness. Many sources now recognise SMI as ‘bipolar, schizophrenia, and other psychoses’. There is a mixture of different illnesses within the patients with a lack of justification and links between the them.

3. Results subtheme 1 (line 295-316): It would be beneficial to see some more quotes here to justify the descriptions.

4. Results subtheme 2 (lines 320-334): It would be good to see a quote/quotes, specifically from the patients here.

5. Results subtheme 3 (lines 354-406): lacking any quotes to justify text.

6. Strengths and limitations (line 556): ‘We chose not to investigate patients with ADHD or dementia, as they have very specialized trajectories that did not align with the aim of this study” – which trajectories are those and how do they differ from the trajectories of those you have included?

7. Conclusion (lines 561-563): very short, would be good to highlight the main recommendations you make here.

Overall, a very interesting article, however, the article goes back and forth between mental illness and severe mental illness. I think this due to not having a solid explanation as to what severe mental illness is and its definition.

Reviewer #2: The author claims that he has done the study with a hermeneutic approach! Unfortunately this claim has not been confirmed in any part of the study.

The author has only conducted a qualitative content analysis. You can rewrite the study in the form of qualitative content analysis.

6. PLOS authors have the option to publish the peer review history of their article (what does this mean?). If published, this will include your full peer review and any attached files.

Reviewer #1: No

Reviewer #2: No

---

## [Author Response · Author response to Decision Letter 0]

7 Sep 2024

Rebuttal letter

Author comments are in italics.

PONE-D-24-23981

”I just felt that everything came tumbling down around me” - Barriers in cancer care for patients with severe mental illness: A qualitative study

PLOS ONE

Dear Editor,

We appreciate the opportunity to respond to the comments provided by the academic editor and the two reviewers. Below, we have addressed each point raised, providing detailed responses to the questions and suggestions. We are grateful for the reviewers' insightful feedback, which has significantly contributed to the improvement of our manuscript.

We hope that the revised manuscript now meets the criteria for publication in PLOS ONE. Please let us know if any additional changes are required.

On behalf of all the authors,

Lars Henrik Jensen

ACADEMIC EDITOR: 

This manuscript has merit and would be of great interest to readers. There are few areas that require further clarification before it can be considered for acceptance:

• Reviewer 2 has queried regarding the 'hermeneutical approach' employed in this manuscript. It would be advisable to explicitly state the characteristics of hermeneutical approach in the methods. I do note that the manuscript has reported the application of reflexivity and interpretive analysis. Rephrasing the methods to be more explicit regarding the hermeneutical approach may be helpful.

We appreciate the comment and have revised the Methods section accordingly. Please refer to lines 94-104. Additionally, we have included two references from Gadamer and Lindseth. Please also see the comment to Reviewer 2 below.

• The definition of severe mental illness was provided in lines 97-99. As mentioned by Reviewer 1, the application of this definition of severe mental illness would require clarification. The participants of the study consisted of patients with mood disorder, anxiety disorder and one patient with psychotic disorder. All these could have a range of mild to severe illness, as patients with severe mood disorder may also have psychotic symptoms. Elaboration of the severity of these diagnoses may help to justify the definition for severe mental illness. 

We recognize that the reviewer’s comments indicate some uncertainty regarding the distinction between "severe mental illness" and "mental illness" within the context of our study. It has become clear that there is some complexity in how these terms are applied. The references cited in our manuscript are partially derived from studies involving patients with SMI, which typically refers to individuals with diagnoses such as schizophrenia and affective disorders. The reviewer is correct in pointing out that our study also includes other diagnostic groups, which may contribute to the confusion, particularly given the presence of multiple diagnoses among participants. This raises questions about whether these are primary and secondary diagnoses, and if so, it indeed becomes challenging to understand the exact composition of our participant group. It is crucial to clarify that all participating patients were recruited from secondary psychiatric services, meaning they were either inpatients or receiving outpatient treatment. Therefore, they all have a history of prolonged and complex engagement with psychiatric care. After the WHO definition, we have added ‘In line with this definition, participants in this study were recruited from secondary psychiatric services, including both inpatient and outpatient settings, with the additional criterion that they had a history of prolonged and complex engagement with psychiatric care.’ (line 117)

• Additional quotes to support the interpretation of results as mentioned by Reviewer 1 is required.

Please see below, Reviewer 1.

• In line with hermeneutical approach, providing more context would be helpful. For example, which patients had undergone ECT? What cancer phase were the participants in at the time of the study? Had they undergone surgery, chemotherapy, or radiotherapy? 

Thank you for the feedback. We have provided additional context to align with the hermeneutical approach. Specifically, we have detailed which patients had undergone ECT, clarified the cancer phases of the participants at the time of the study, and outlined their prior treatments, including surgery, chemotherapy, or radiotherapy. This will enhance understanding of the participants' treatment trajectories and overall experiences.

In ‘Participants’:

“During the study, three patients were undergoing curative or adjuvant oncology treatment, while one patient was receiving life-prolonging medical oncology treatment. Another patient, who had been discontinued from adjuvant oncology treatment, was instead being monitored with control CT scans at the time of their inclusion in the study. This patient was still considered able to provide valuable information regarding the treatment trajectory and was therefore retained in the study.

Three patients were undergoing radiotherapy, and three patients (with two overlapping with radiotherapy) were receiving medical oncology treatments, administered either as injections or oral tablets. Additionally, three patients had previously undergone surgical removal of their cancer, and two had received intravenous chemotherapy before the study began.

Two patients were undergoing electroconvulsive treatment (ECT), and all patients were receiving psychopharmacological treatment.”

• Why did the study specifically include the role of GPs as it was not evident in the objectives?

Thank you for raising this point. We recognize that the initial manuscript did not provide sufficient background on the role of general practitioners (GPs) in Denmark. To address this, we have now added the following statement to the Methods section: “In the Danish healthcare system, the general practitioner (GP) plays a central role for each patient, serving as the primary care provider, managing overall health, and coordinating care across sectors.” (line 149). This addition clarifies the integral role of GPs in the Danish context, which justifies their inclusion in the study, even though it was not explicitly mentioned in the original objectives. The involvement of GPs is essential for understanding the full spectrum of care provided to patients, especially those with complex health needs across sectors.

Reviewers' comments:

Comments to the Author

Reviewer #1: Thank you very much for the opportunity to review this article. Overall, it is a very good piece and it is great to see more work is being done in this area. I have a few notes (please see below).

1. Introduction: you mention both mental illness and severe mental illness (SMI). SMI has been referred to as schizophrenia and bipolar, however, no examples have been given for mental illness – should all of this say severe mental illness? Also, more background could be given on what severe mental illness is. Such as what is schizophrenia/bipolar etc and how many people are affected by it? Are there any other severe mental illnesses?

2. Methods: It is not clear if the patients were diagnosed with a SMI or a mental illness. Many sources now recognise SMI as ‘bipolar, schizophrenia, and other psychoses’. There is a mixture of different illnesses within the patients with a lack of justification and links between the them.

Points 1 and 2, please see the answer above about the definition of severe mental illness. 

3. Results subtheme 1 (line 295-316): It would be beneficial to see some more quotes here to justify the descriptions.

4. Results subtheme 2 (lines 320-334): It would be good to see a quote/quotes, specifically from the patients here.

5. Results subtheme 3 (lines 354-406): lacking any quotes to justify text.

Points 3, 4 and 5, thank you for the opportunity to elaborate and further support our analysis. In response to your comments, we have added eight quotes to provide additional justification for our descriptions. Please see in the manuscript, Subtheme 1, 2 and 3.

6. Strengths and limitations (line 556): ‘We chose not to investigate patients with ADHD or dementia, as they have very specialized trajectories that did not align with the aim of this study” – which trajectories are those and how do they differ from the trajectories of those you have included?

Thank you for your question regarding the exclusion of patients with ADHD and dementia from our study. The central reason for this decision is that ADHD and dementia did not meet our criterion of prolonged and complex engagement with secondary psychiatric care, which was a key focus of our study. The reason is now specified in the Strengths and Limitations section. We hope this clarification addresses your concern.

7. Conclusion (lines 561-563): very short, would be good to highlight the main recommendations you make here.

We appreciate the reviewer’s suggestion to expand the conclusion. In response, we have revised the conclusion to offer a more detailed summary of our key findings and to clearly articulate the main recommendations stemming from our study.

“Conclusion

This study underscores the complexity of care for patients with cancer who also suffer from severe mental illness, revealing key areas for improvement. Our analysis identified the overarching theme of "Complexity on many levels," with four subthemes: the impact of the cancer trajectory on mental illness, the multifaceted vulnerability of patients, the fragmentation and lack of structure in the healthcare system, and the critical role of the patient-health professional relationship. Barriers to effective care included the exacerbation of mental illness due to the cancer trajectory and systemic fragmentation. Facilitators included recognizing and leveraging the patient's own resources and adopting a person-centered approach to care. By addressing these barriers and enhancing these facilitators, healthcare providers can better meet the needs of this vulnerable population.”

Overall, a very interesting article, however, the article goes back and forth between mental illness and severe mental illness. I think this due to not having a solid explanation as to what severe mental illness is and its definition.

Reviewer #2: The author claims that he has done the study with a hermeneutic approach! Unfortunately this claim has not been confirmed in any part of the study.

The author has only conducted a qualitative content analysis. You can rewrite the study in the form of qualitative content analysis.

We appreciate the concern regarding the use of the term "hermeneutic approach" in our manuscript. After careful consideration, we respectfully maintain that the study is conducted within a hermeneutic framework. To address the reviewer's concern and to clarify our approach, we have expanded the Methods section to provide a more detailed explanation of the hermeneutic principles guiding our analysis. 

These revisions include additional background on the hermeneutic methodology and a more explicit description of how this approach was applied throughout the study. We believe these enhancements will make the hermeneutic nature of our analysis clearer.

END ==============================

---

## [Decision Letter · Decision Letter 1]

8 Nov 2024

”I just felt that everything came tumbling down around me” - Barriers in cancer care for patients with severe mental illness: A qualitative study

PONE-D-24-23981R1

Dear Dr. Jensen,

We’re pleased to inform you that your manuscript has been judged scientifically suitable for publication and will be formally accepted for publication once it meets all outstanding technical requirements.

Although reviewer 2 has suggested some further revisions to the manuscript, I have read the manuscript completely and am satisfied that points raised during the previous round of review have been satisfactorily addressed. As the manuscript is not a methodological discussion, I felt that further revisions are no longer required. The current discussions are good and relevant to the objectives of the study. Thank you for contributing new knowledge regarding care for this vulnerable population. 

Kind regards,

Chai-Eng Tan

Academic Editor

PLOS ONE

Additional Editor Comments (optional):

Reviewers' comments:

Reviewer's Responses to Questions

**Comments to the Author**

1. If the authors have adequately addressed your comments raised in a previous round of review and you feel that this manuscript is now acceptable for publication, you may indicate that here to bypass the “Comments to the Author” section, enter your conflict of interest statement in the “Confidential to Editor” section, and submit your "Accept" recommendation.

Reviewer #1: All comments have been addressed

Reviewer #2: (No Response)

2. Is the manuscript technically sound, and do the data support the conclusions?

Reviewer #1: Yes

Reviewer #2: (No Response)

3. Has the statistical analysis been performed appropriately and rigorously? 

Reviewer #1: Yes

Reviewer #2: (No Response)

4. Have the authors made all data underlying the findings in their manuscript fully available?

Reviewer #1: Yes

Reviewer #2: (No Response)

5. Is the manuscript presented in an intelligible fashion and written in standard English?

Reviewer #1: Yes

Reviewer #2: (No Response)

6. Review Comments to the Author

Reviewer #1: The authors addressed all comments made and therefore, I have accepted the revisions. Thank you very much.

Reviewer #2: At the end of the introduction, the researcher's personal experience of the phenomenon under study should be mentioned. For what purpose did you use the hermeneutic approach in this study?

Why did you use Gadamerian hermeneutics? Why didn't you use Heidegger's hermeneutics? What is the difference between these two approaches?

Rewrite the entire discussion with a Gadamerian hermeneutic approach.

7. PLOS authors have the option to publish the peer review history of their article (what does this mean?). If published, this will include your full peer review and any attached files.

Reviewer #1: No

Reviewer #2: No

---

## [Editor Report · Acceptance letter]

14 Nov 2024

PONE-D-24-23981R1 

PLOS ONE

Dear Dr. Jensen, 

I'm pleased to inform you that your manuscript has been deemed suitable for publication in PLOS ONE. Congratulations! Your manuscript is now being handed over to our production team.

Kind regards, 

on behalf of

Dr. Chai-Eng Tan 

Academic Editor

PLOS ONE